# Validity of DSM-5 Oppositional Defiant Disorder Symptoms in Children with Intellectual Disability

**DOI:** 10.3390/ijerph18041977

**Published:** 2021-02-18

**Authors:** Victor B. Arias, Virginia Aguayo, Patricia Navas

**Affiliations:** 1Institute on Community Integration, University of Salamanca, 37005 Salamanca, Spain; aguayo@usal.es (V.A.); patricianavas@usal.es (P.N.); 2Department of Personality, Assessment and Psychological Treatments, University of Salamanca, 37005 Salamanca, Spain

**Keywords:** oppositional defiant disorder, dual diagnosis, intellectual disabilities, diagnostic overshadowing, dual diagnosis, DSM-5

## Abstract

Oppositional defiant disorder (ODD) is one of the most frequently diagnosed disorders in children with intellectual disabilities (ID). However, the high variability of results in prevalence studies suggests problems that should be investigated further, such as the possible overlap between some ODD symptoms and challenging behaviors that are especially prevalent in children with ID. The study aimed to investigate whether there are differences in the functioning of ODD symptoms between children with (*n* = 189) and without (*n* = 474) intellectual disabilities. To do so, we analyzed the extent to which parental ratings on DSM-5 ODD symptoms were metrically invariant between groups using models based on item response theory. The results indicated that two symptoms were non-invariant, with degrees of bias ranging from moderately high (“annoys others on purpose”) to moderately low (“argues with adults”). Caution is advised in the use of these symptoms for the assessment and diagnosis of ODD in children with ID. Once the bias was controlled, the measurement model suggested prevalences of 8.4% (children with ID) and 3% (typically developing children). Theoretical and practical implications are discussed.

## 1. Introduction

Oppositional defiant disorder (ODD) is characterized by a pattern of anger/irritability, argumentative/defiant behavior, or vindictive behavior exhibited during interactions with at least one individual other than a sibling. The prevalence of ODD in the child and adolescent population has been estimated at between 1% and 11% (mean 3.3%), depending on the study and age onset [1].

ODD is one of the most frequently diagnosed disorders in children with intellectual disabilities (ID) [2,3,4]. In their review of the prevalence of chronic health conditions in children with ID, Oeseburg et al. (2011) estimated a weighted mean ODD prevalence of 12.4% (CI 10.7–14.4%) [4]. Other studies have estimated the prevalence (or the percentage of scores in a clinical range): 13.3% in children with ID (versus 2.3% in typically developing children (TD) [3]), 23.1% (versus 5.2% in TD children [5] (the average comorbidity observed between ADHD and ODD for the 5–8-year age range)), 49% in children with ID and attention-deficit/hyperactivity disorder (versus 3.4% in children with ID only [6]), 49% (versus 18% in TD children [7]), 13.9% [2], and from 34.7% to 47.8% when depending on age [8]. Overall, the prevalence ratios of ODD between children with ID and TD children range from 2–5 to 1–5 [9]. The relative risk ratios of children with ID in comparison to their TD peers vary from 1.60:1 [10] to 1.70:1 [11].

The high variability of results suggests problems that should be further investigated, such as the impact of age, small samples, the use of different measuring instruments and diagnostic procedures, and the use of diagnostic algorithms that were initially designed for children without ID (e.g., those based on the DSM). Furthermore, given the possible overlap between some ODD symptoms and the challenging behaviors that are especially prevalent in children with ID [8], the validity of behavioral indicators needs to be thoroughly analyzed.

### 1.1. Validity of the ODD Diagnosis in Children with Intellectual Disabilities

The validity of the ODD diagnosis in children with ID is still uncertain [8]. Beyond prevalence, most specific studies have researched ODD characteristics exclusively in children with autism spectrum disorders, because its core symptoms and comorbidity pose a challenge for differential diagnosis [12,13,14,15,16]. However, there are very few studies devoted to comparing ODD between children with ID and TD children. Christensen, Baker, and Blacher (2013) examined the epidemiology of ODD by comparing samples of children between ages 5 and 9 with ID (*n* = 49), borderline intellectual functioning, and TD [11]. They found that children with ID consistently showed higher rates of ODD than TD children. They found no group differences in gender distribution, age onset, or inter-annual stability of the disorder. In addition, they also observed no substantial discrepancies in how children with and without ID met the diagnostic criteria for ODD, with no significant differences in the raw frequency with which each symptom was endorsed.

The assessment of ODD in children with ID involves potential difficulties that require further research. First, the increased prevalence of challenging behaviors (CB) in children with ID is well known [17,18,19,20]. Since a higher rate of CB appears to be normative in children with ID [8], the overlap between such problems and ODD symptoms may make it difficult to detect and diagnose the disorder, even leading to confusion between ODD symptoms and ID epiphenomena [21]. This problem seems especially relevant for ODD symptoms, which overlap with certain challenging behaviors (e.g., the defiant facet of ODD and CB related to disruption and noncompliance). Second, it is unknown to what extent discrepancies between prevalence estimates are due to the use of different measurement instruments with diverse sets of symptoms and different scoring procedures. Lacking a properly validated set of ODD symptoms for the ID population hampers both establishing appropriate diagnostic baselines and reaching consistent conclusions about the prevalence of the disorder. To ensure the validity of prevalence rates, it is necessary to reach a consensus on the classification framework [4].

### 1.2. The Present Study

One way to investigate the validity of the ODD diagnosis is to compare the presentation of the disorder between children with and without ID to determine if the same diagnostic entity is being captured in both groups [8]. However, for comparisons to be meaningful, the symptoms used to assess ODD must work the same way in both groups. Failure to demonstrate metric equivalence would mean that, in children with ID, the occurrence of symptoms is partly determined by variables other than the presence of ODD. This may lead to three negative consequences: (a) the difficulty or even impossibility of separating symptoms due to ODD from other problematic features, such as CB, together with problems in providing accurate prevalence estimates; (b) the impossibility of making meaningful comparisons of ODD ratings between children with and without ID; and (c) the potentially serious risks for the diagnosis, given the increased likelihood of false positives or negatives due to measurement bias.

The study aimed to investigate whether there are differences in the functioning of ODD symptoms between children with and without ID. To do so, we analyzed the extent to which parental ratings on DSM-5 ODD symptoms were metrically invariant between the two groups. Once the problematic symptoms were detected, we estimated the size of the bias and its impact on the assessment of ODD. Finally, from the best-fitting model, we estimated the latent scores of both groups in order to compare their distributional properties in a common metric with controlled bias. A bias-controlled, common metric for ODD symptoms could be useful for the better assessment and diagnosis of ODD in children with ID. Finally, an assessment of the measurement invariance of ODD symptoms also informs us whether all symptoms or only a subset are appropriate for the assessment of ODD in children with ID.

## 2. Materials and Methods

### 2.1. Participants

Two regular schools and three special education schools were contacted to facilitate the participants’ recruitment. After their approval to collaborate, meetings were arranged with scholars’ parents. In the first contact, parents were introduced to the study details and the main objectives of the research. Those who agreed to collaborate voluntarily and free of charge gave their written informed consent, and meetings with the research team were organized to fulfill the ODD scales. The scales were completed in pencil and paper format under an investigator’s supervision so that no data were missing. The measurement forms were stored, guaranteeing the confidentiality and anonymity of the participants. The research was conducted following the principles of the 1964 Helsinki Declaration and its later amendments, as well as the ethical standards of the institutional and national research committees that supported this study.

Information was recorded on 663 children (ages 7–15). They were divided into two groups: sample 1 was composed of 474 TD children (42% males; mean age = 10.3, SD = 2.3), and sample 2 comprised 189 children with a formal diagnosis of ID made by specialized clinicians (46% males; mean age = 10.8, SD = 2.2). The formal diagnosis of ID was characterized as mild (64%) or moderate (36%), and medical records displayed additional conditions in 65% of the cases, including Down syndrome (12.7%), attention-deficit hyperactivity disorder (ADHD; 10.6%), behavioral problems with unspecified diagnosis (9.5%), cerebral palsy (6.3%), autism spectrum disorder (3.2%), severe health problems (2.6%), epilepsy (4.8%), or other (specific learning disorder, fragile X syndrome, Williams syndrome, specific language disorder, and mental disorder). The need for support was gathered in 84.1% of children with ID, varying between limited support (23.8% of the children), intermittent support (33.3%), and extensive support (27%). Concerning schooling, 79.4% of children with ID attended special education schools, while 20.6% were present at regular schools within special education programs.

### 2.2. Measure

Parents completed the ODD subscale of the Child and Adolescent Behavior Inventory (CABI [22]), whose adaptation to Spanish has been previously tested [23,24,25]. The ODD subscale comprises eight items referring to recurrent behaviors that make up the criteria symptoms in the DSM-5 [1]. The items are (a) argues with adults, (b) loses temper, (c) refuses to comply with requests or rules, (d) annoys others on purpose, (e) blames others for his or her mistakes or misbehavior, (f) is often susceptible or easily annoyed, (g) is spiteful or vindictive, and (h) appears often angry and resentful. Each of the elements is rated on a 6-point occurrence scale (1—rare (never or about once per month), 2—seldom (about once per week), 3—sometimes (several times per week), 4—often (about once per day), 5—very often (several times per day), and 6—almost always (many times per day)).

### 2.3. Data Analysis

#### 2.3.1. Analytical Approach

In this study, we used the graded response model (GRM [26]) implemented in IRTPRO v. 4.0 [27]. The GRM assumes, in addition to the usual item response theory assumptions, that the categories in which the individual is scored can be ordered or hierarchized, as in the case of probabilistic scales of summative estimates. The model is intended to obtain more information than if there were only two response levels (e.g., “yes”–“no”), so it is an extension of the two-parameter logistic model (2P-LM) to ordered polytomous categories. The GRM specifies the probability of a person being scored in an i*k* or higher category as opposed to being scored with a lower category when the scoring system has at least three categories, depending on the estimated trait level for each individual.

Prior to the estimation of the GRM models, unidimensionality and local independence were verified through an optimized parallel analysis [28] on the matrix of polychoric correlations, and the inspection of the χ2 values of the matrix of expected and observed frequencies in the response to each item (χ2 values greater than 10 suggest a substantial violation of local independence [29]).

#### 2.3.2. Differential Item Functioning Analysis (DIF)

Firstly, the quality of the measurement model was evaluated by inspecting the mean of the standard errors of the discrimination and location parameters. Values below 0.20 suggest that the parameters have been estimated with very good accuracy; between 0.20 and 0.30 means good; between 0.30 and 0.40, normal; and above 0.40, poor accuracy [30].

Secondly, the statistical significance of the differences in the groups’ parameters was estimated using Wald’s test [31]. Wald’s test consists of an iterative analysis where all items are treated as DIF possibilities in the first instance. In a second round, a new model is specified in which the symptoms that did not show significant DIF in the previous step are used as anchors to re-estimate the DIF of the items that remain as possibilities. This second step is repeated until a stable set of non-invariant items is obtained. To decide which symptoms were affected by the DIF, a probability of less than 1% was used as a criterion to determine whether the differences between parameters observed were due to random fluctuations in the data (we chose a conservative confidence level in order to avoid detecting effects with negligible size in practice).

#### 2.3.3. Analysis of the Impact of Differential Symptom Functioning

In this step, we inspected the size of the DIF in the affected items, as well as the area of the latent variable with the greatest amount of bias in the cases of non-monotonic DIF. The expected standardized score difference (ESSD [32]) was used as an estimator of the total DIF size. Since ESSD is expressed in the same metric as the latent variable, it usually acquires values between ±3, and its absolute value can be interpreted as a Cohen *d* [33]. The interpretation of the ESSD was supported by the visual inspection of the characteristic curves of the items affected by DIF.

## 3. Results

### 3.1. Unidimensionality and Local Independence

Figure 1 shows the results of the parallel analysis. From the second component, the 95th percentile of the variance explained by the simulated matrices (1000 permutations) exceeds those generated by the data. This finding suggests the presence of a preponderant factor in both samples (ID and TD), which in turn suggests that ODD can be measured as a unidimensional variable in both groups. The LDχ2 values of the matrix of expected and observed frequencies in the response to each item were in all cases less than 10 in sample 1 and greater than 10 in one of the 28 contrasts in sample 2. No clusters of high LDχ2 values were observed that would lead to suspicion of relevant residual systematic variance.

### 3.2. Analysis of the Differential Functioning of the Symptom

Table 1 shows the results of Wald’s test. In the first iteration, two items (“argues with adults” and “annoys others on purpose”) showed significant chi-square values, a result that persisted in the second iteration.

The latent mean estimated by the partially invariant model for children with ID was 0.37 standard deviations higher than that estimated for TD children. The mean of the standard errors of estimation was 0.19 (SD = 0.03) for the discrimination parameters and 0.11 (SD = 0.04) for the localization parameters. According to the classification proposed by Tay, Meade, and Cao (2015), it can be concluded that the estimation precision is very high [30], even though the focal sample is relatively small (useful *n* = 189).

### 3.3. Analysis of the Size and Impact of Differential Item Functioning

Figure 2 and Figure 3 represent the probability of endorsing the CABI’s category 4 (i.e., the ODD symptoms occur often or about once per day) or higher, conditionally to the ODD level (theta), for items affected by DIF: “annoys others on purpose” (Figure 2) and “argues with adults” (Figure 3).

The item “annoys others on purpose” presented moderately small DIF if we consider the whole latent continuum (ESSD = 0.33) and moderately high (ESSD = 0.66) if we consider only the theta region most relevant for diagnosis (from two standard deviations above the mean). This result implies that, at high levels of the variable, the score of the ID group was on average 0.66 standard deviations lower than that of the TD group, conditionally on the ODD level. Thus, for example, in order to have a 50% probability of endorsing the symptom (responding in the category “often” or higher; [34]), a child with ID must have an ODD (theta) level of approximately 1.2 standard deviations above the mean, while a TD child requires an ODD level of two or more standard deviations. These results suggest the presence of a significant amount of bias that cannot be ignored and is concentrated at clinical ranges.

The second affected item (“argues with adults”) presented DIF with a small effect size along the entire latent continuum (ESSD = 0.19), and it was somewhat larger when considering only the range of theta potentially relevant to the diagnosis (ESSD = 0.37). The direction of the bias was opposite to that of the previously described item. In this case, TD children showed a systematically higher probability of endorsing the symptom, regardless of the ODD level.

### 3.4. Distribution of Latent Oppositional Defiant Disorder Scores

Figure 4 shows the distribution of latent scores (M = 0, SD = 1) estimated from the partially invariant model. The form of the distribution was similar between groups, with a shift towards higher levels of the variable in the ID sample (a mean difference of 0.37 standard deviations). Both distributions showed a clear positive asymmetry, a result expected from the application of clinical scales to the general population [35]. Possibly the most noticeable difference was the higher density of children with ID with very high levels of ODD (>2 SD). The proportion of children with ID at levels for potential diagnosis (+2 SD and above) was 8.4%, compared with 3% of TD children.

## 4. Discussion

This study was designed to analyze measurement invariance of ODD symptoms gathered from parent ratings on a scale based on DSM-5 symptoms of typically developing (TD) children and children with mild or moderate intellectual disabilities (ID). Once the symptoms with significant differential functioning were identified, the impact of the bias on the ODD measure was assessed by estimating the size of the bias at the individual symptom level.

Two of the eight ODD symptoms were suspected of differential functioning. Firstly, the symptom “annoys others on purpose” showed approximately uniform DIF, with the probability of endorsing the item consistently higher in children with ID, regardless of ODD level. The effect size of the DIF was moderately high at diagnosis-relevant levels (i.e., more than 2 SD above the mean). A plausible explanation of this result is that the symptom rating may be affected by the presence of behaviors unrelated to ODD but topologically similar to the symptom, such as challenging-disruptive behaviors. The size and direction of the bias warns of a high chance of false positives, so it is advisable to use this symptom with caution and to consider it for diagnosis only when other symptoms are present and the problematic behavior is not better explained by other causes outside the ODD.

Secondly, the symptom “argues with adults” displayed uniform DIF. The probability of endorsing the item was consistently higher in TD children, regardless of the ODD level. The effect size of the DIF was low, so it did not acquire enough magnitude to substantially bias the measure of the disorder, and, therefore, it does not seem urgent to modify or eliminate it. Nevertheless, note that “argues with adults” is the ODD symptom with the strongest verbal component, since this behavior requires a certain level of expressive language skills. It is plausible that the difficulties in language development, frequent in children with ID [36], alter the relationship between the symptom and the underlying trait, resulting in a lower probability of endorsement. In this way, a deficit in verbal expressive skills could be altering the behavioral manifestation of the symptom and therefore hindering its detection in those cases in which it is present. It is likely that in these cases (i.e., cases with insufficient verbal skills to argue with the adult, or even absence of verbal language), the symptom manifests itself in alternative ways (e.g., through non-verbal defiant behaviors). Even though the size of the bias is low, it is advisable to use that symptom with caution and to consider it for diagnosis only after ensuring that the current development of the child’s verbal skills is not interfering with the scoring of the symptom.

In terms of measurement of the disorder (expressed through ODD latent scores), the same symptoms have been calibrated in samples with and without ID, so it was possible to evaluate their ODD level on a common scale, after controlling for measurement error effects and possible bias due to the specific characteristics of each sample. In this study, we found generally higher scores in children with ID. This result was expected, given the role of ID as a risk factor in the development of mental disorders [37,38] and the available results on the prevalence of ODD [4,8]. However, this result does not imply that ID necessarily entails relevant oppositional/defiant behavior problems, as shown by the presence of a considerable proportion of children with ID in the middle and lower ranges of the variable. Although the estimated prevalence in TD children (3%) was very similar to those suggested by previous prevalence studies in the general population [1], the prevalence in children with ID (8.4%) was substantially lower than that reported by most previous studies. The reasons for this discrepancy could be explained by the defining characteristics of the samples used, as well as in the establishment of the cut-off point to determine the presence of ODD reported in the different studies.

## 5. Conclusions, Limitations, and Future Directions

An important limitation of this study is the sample size of children with ID. While it has allowed for the precise estimation of model parameters (according to the standard errors of the discrimination and location indices), the limited sample size has prevented deeper analyses including relevant grouping variables, such as age ranges, the child’s developmental level, or the potential effect of behavioral phenotypes with specific characteristics [39]. On the other hand, future studies must check whether the results presented here are extensible to other informants, especially teachers and clinicians.

The results of this study suggest that the ODD symptoms of DSM-5 may be valid for the assessment of the disorder in children with ID, provided they are used with appropriate caution. Demonstration of sufficient invariance suggests that the same construct is being captured in children with TD and ID and that a common metric could be used for the compared groups [40,41]. However, metric equivalence, while a core requirement, does not ensure itself the diagnostic validity of symptoms in a specific population. In turn, it is necessary to assess the convergence of other elements, such as clinical descriptions, family history, aspects related to differential diagnosis, etiological factors, age and developmental course, and various outcomes [42,43]. There is also a need to investigate the causes of heterogeneity in prevalence estimates across studies, for example, by exploring potential moderators that may affect population prevalence estimates in children with ID. Despite these limitations, we believe that this study contributes to alleviating the lack of comparison groups as one of the main shortcomings in the research of behavioral disorders in children with ID [8] and adds knowledge to a still underexplored problem.

## Figures and Tables

**Figure 1 ijerph-18-01977-f001:**
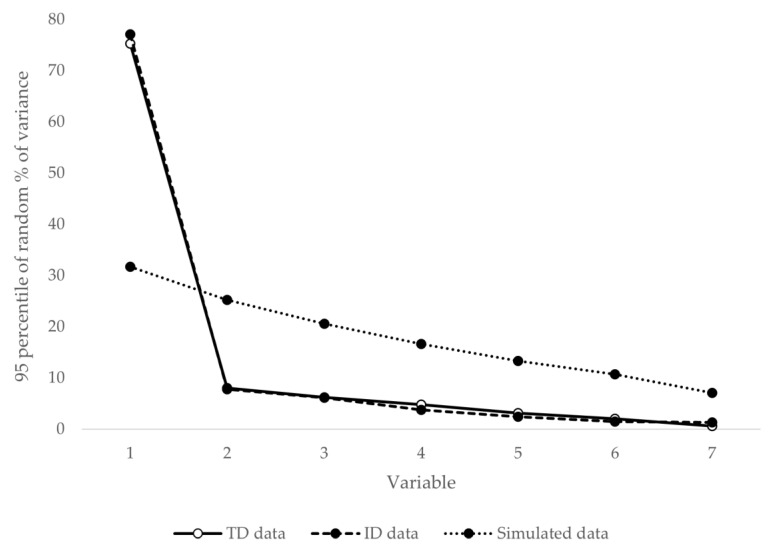
Results of the parallel analysis. Note: TD = typical development; ID = intellectual disability group; variable = number of common factors.

**Figure 2 ijerph-18-01977-f002:**
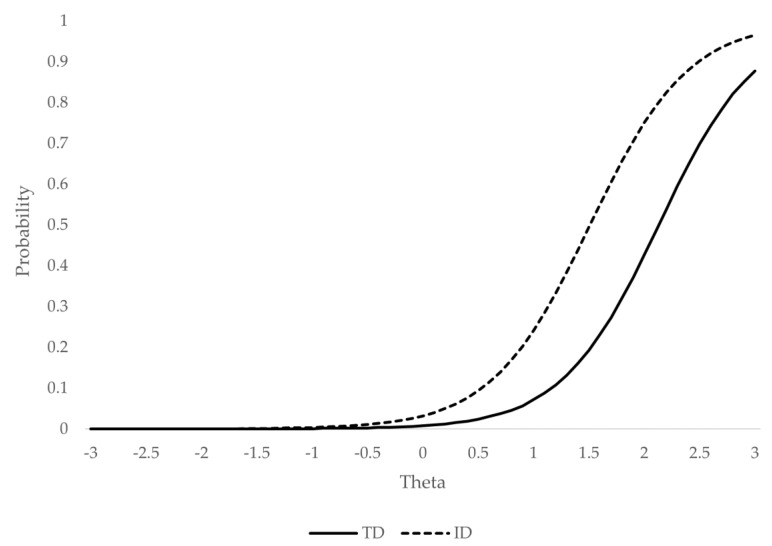
Probability of endorsing the category “often” or higher for the symptom “annoys others on purpose”. Note: TD = typical development; ID = intellectual disability; theta = standardized ODD latent score (M = 0; SD = 1).

**Figure 3 ijerph-18-01977-f003:**
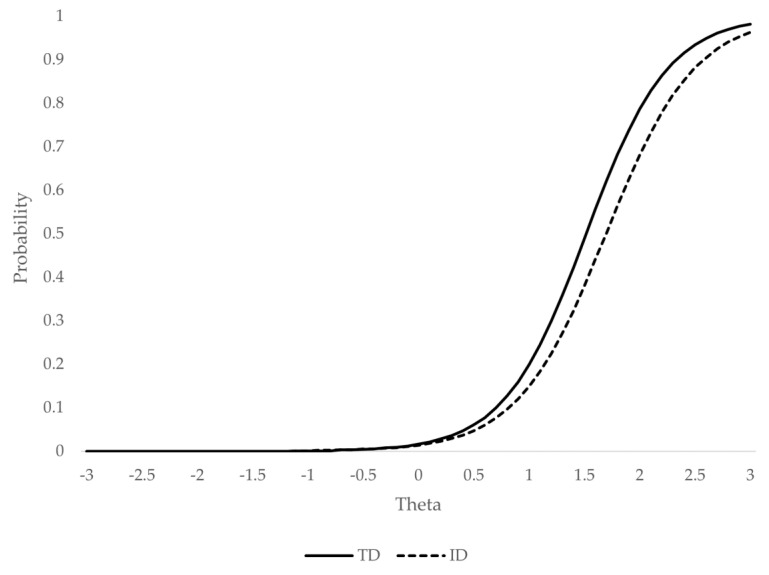
Probability of endorsing the category “often” or higher for the symptom “argues with adults”. Note: TD = typical development; ID = intellectual disability.

**Figure 4 ijerph-18-01977-f004:**
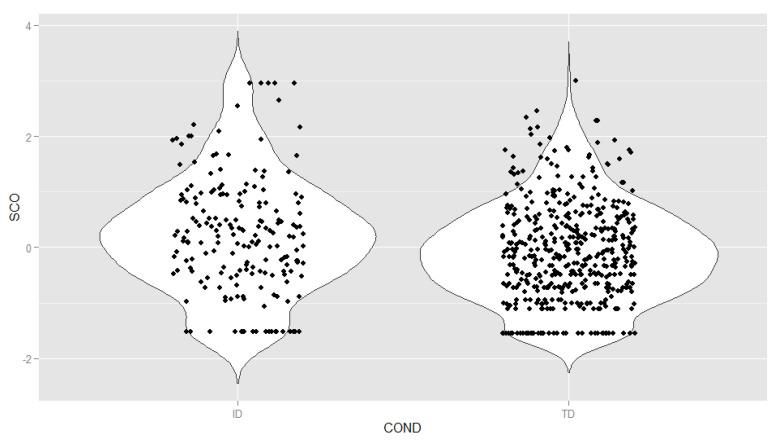
Distribution by group of the latent oppositional defiant disorder scores (M = 0, SD = 1). Note: TD = typical development; ID = intellectual disability group; SCO = standardized ODD latent scores; COND = condition.

**Table 1 ijerph-18-01977-t001:** Differential item functioning analysis of oppositional defiant disorder (ODD) symptoms.

Symptom ^1^	Iteration 1		Iteration 2	
	Wald χ2	*p*	Wald χ2	*p*
**Argues with adults**	18.1	0.006	20.9	0.002
Loses temper with others	12.6	0.051	0	1
Actively defies or refuses to obey adults’ requests or rules	12.3	0.055	0	1
**Annoys others on purpose**	20.8	0.002	21.9	0.001
Blames others for his or her mistakes or misbehavior	3.1	0.797	0	1
Becomes annoyed or irritated by the behavior of others	9	0.173	0	1
Appears angry or resentful toward others	7.4	0.290	0	1
Spiteful or vindictive toward others	3.6	0.733	0	1

^1^ In bold = items with significant differential item functioning (*p* < 0.01).

## Data Availability

The data presented in this study are available on request from the corresponding author

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
