# Peer review of "Validity of DSM-5 Oppositional Defiant Disorder Symptoms in Children with Intellectual Disability"

_ijerph, 2021, doi:10.3390/ijerph18041977_

Round 1
Reviewer 1 Report
The manuscript entitled “Validity of DSM-5 oppositional defiant disorder symptoms in children with intellectual disability: describes the comparison between children with ODD can be sub-grouped into ID and TD. There is a bias in two measurements viz., annoys others on purpose to argues with adults.
Comments
- The manuscript is well written, however, few points need to be addressed to improve the concept. How do the authors differentiate between TD – typically developing child, to NT-Neurotypical child/children? If it is the same can they change TD to NT, which is more conventional?
- There is always a spectrum or range within any group of diagnoses. Did the authors test the normal children who adapt to the norms of the society, they are referred to as NT (neurotypical children or previously, typically developing children)?If the authors are subgrouping ODD children, how would they compare them with normal healthy children? If TD grouped under ODD, the authors have to describe them differently.
- Figure 1 The results of the parallel analysis. Please describe the graph in detail and provide an inset graph with variable 2-7 because the changes are seen. The abscissa and ordinance should specify the variable. Current presentation the observation is skewed due to the 1st variance, hence the difference seen from 2-7 becomes very subtle. Is there a way to select from 2 to 7 and show the difference?
- With regard to the data presented in table 1: Some of the symptoms described can be questioned such as “Argues with adults” Some of the listed symptoms can be questioned. This depends on the age and the situation, how they are distracted. Is it possible to correlate with the age and gender of the children?
- The above question was prior to reading the discussion. The authors have described caution in measuring the above-mentioned criteria. It would be helpful to narrate why such measurements can be an oddity?
- For all the graphical representation, the x-axis and the y axis needs to be well defined and not abbreviated. Please provide a sentence or two in figure legend to describe what the readers should look for in the graph because the figures are not well explained in the text.
- While the measurement of annoyance equates to verbal communication skills, it would be nice to correlate the verbal skills to behavioral skills. Was any measurement done on assessing the verbal skills?
Minor comments
- Abstract – 3rd sentence - change:‘The aim of this study was to investigate’ to ‘The study aimed at’
- 1.1 second paragraph last two sentences – “ID population hamper to both establish” – delete “to”
- 1.1 second paragraph last line – “it is necessary to previously reach consensus on” – “reach a consensus”
- 1.2 second paragraph – “The aim of this study was to investigate” – it is better to say “The study aimed at”
- 2.2 Measure – “(1 - almost never” – category can be “rare”
- 2.3.1 second paragraph – “matrix of polycoric” – it is polychoric
- 2.3.2 last line of the first paragraph – “between .20 and .30 mean good;” – change mean to “means”
- 2.3.3. third line of the first paragraph – “was used as estimator of” – change to “as an estimator”
- 3.3 last paragraph – “showed systematically higher” – change to “showed a systematically”
- Discussion 3rd paragraph – “contribute to alter the relationship” – change to “altering”
- Conclusion – “it is necessary that future studies check” – change to “future studies must check”
- Conclusion – second paragraph – “enough to ensure diagnostic validity“–change to “ensure the diagnostic".
Reviewer 2 Report
The current study explores whether ODD symptomology showed metric invariance between a group of typically developing children and children with intellectual disability. Results suggest that two items in the DSM-V are not metrically invariant between TD and ID children and may be biased and. Given the co-morbidity between ID and ODD diagnoses, this is an important area of research and has the potential to contribute to our understanding of ODD in ID populations as well as inform the application of diagnostic tools of OD to ID populations. However, I do have several questions/comments that could further improve the manuscript, which I detail below:
- The current study estimates the prevalence of ODD to be 8.4% in ID children and 3% of TD children using latent scores from the partially invariant model. Given that the authors make the case that bias of specific ODD symptoms between TD and ID groups could affect prevalence rates of the disorder, I can’t help but wonder how removal of both just the single the symptom “annoys others on purpose” as well as removal of both potentially problematic items compares to use of the full 8 items on prevalence rates. This seems like a question that could be feasibly addressed in the manuscript and would both increase the applicability of the findings and strengthen the authors’ conclusions that these symptoms could bias prevalence rate estimates in ID populations.
- There is significant heterogeneity in the ID subsample both in terms of type of diagnosis (e.g., autism, ADHD, cerebral palsy, Down’s syndrome) and severity. In addition, it appears as though the authors used school classification of “special needs” students, which includes behaviorally challenged students. These students may or may not actually meet diagnostic criteria for ID. These limitations (i.e., heterogeneity of ID type/severity, lack of formal ID diagnosis) should be addressed in the Discussion section.
- Discussion of future directions is limited. In addition to addressing the above limitations, future studies could also attempt to address heterogeneity in prevalence estimates across studies by (a) conducting a meta-analysis of prevalence rates and (b) exploring potential moderators that may impact prevalence rates across studies.
Minor:
- More information on the demographics of the sample (e.g., location of sample, race/ethnicity of sample, gender breakdown, SES status, etc.) would be helpful in contextualizing the research to existing and future data.
- The authors imply that laten scores are “free of bias” (p. 3). While latent analysis can reduce or correct bias, particularly in regard to measurement error, I don’t think it is entirely accurate to say that latent scores are bias free.
- Why was a probability of less than 1% chosen? Was this to account for multiple comparisons/analyses? Given that the standard is usually 5%, an explicit rationale for this decision would be helpful.
- The authors give a very clear and thorough description of the findings related to the item “annoys other people” (p. 6). The description of the results for “argues with adults” is less thorough. I think presenting the results in the same way for both items would be helpful to readers.
